organic chemistry/biochemistry/medicinal chemistry

β-elemene, SeO2, cyclohexyl ring modifications, anti-proliferation

**Authors for correspondence:**
Xiang-Yang Ye
e-mail: xyye@hznu.edu.cn
Tian Xie
e-mail: xbs@hznu.edu.cn

This article has been edited by the Royal Society of Chemistry, including the commissioning, peer review process and editorial aspects up to the point of acceptance.

†These two authors contributed equally.

# β-Elemene derivatives produced from SeO2-mediated oxidation reaction

Xingrui He[1,2,3,4,†], Xiao-Tao Zhuo[1,2,3,†], Yuan Gao[5], Renren Bai[6], Xiang-Yang Ye[1,2,3,7] and Tian Xie[1,2,3,7]

[1]Key Laboratory of Elemene Class Anti-Cancer Chinese Medicine of Zhejiang Province,
[2]Engineering Laboratory of Development and Application of Traditional Chinese Medicine from Zhejiang Province, and [3]Holistic Integrative Pharmacy Institutes (HIPI),
School of Medicine, Hangzhou Normal University, Hangzhou, Zhejiang 311121,
People's Republic of China
[4]School of Pharmacy, Liaocheng University, Shandong 252000, People's Republic of China
[5]School of Clinical Medicine, Guangdong Pharmaceutical University, Guangzhou, Guangdong 510000, People's Republic of China
[6]College of Pharmaceutical Science, Zhejiang University of Technology, Hangzhou, Zhejiang 310014, People's Republic of China
[7]Collaborative Innovation Center of Chinese Medicines from Zhejiang Province, Hangzhou Normal University, Hangzhou, Zhejiang 311121, People's Republic of China

XH, 0000-0002-7591-8022; X-TZ, 0000-0002-4388-4809;
YG, 0000-0003-2254-6381; RB, 0000-0002-3511-5794;
X-YY, 0000-0003-3739-0930; TX, 0000-0001-7066-1443

Herein, we report the first access of β-elemene derivatives through the SeO2-mediated oxidation reaction. Several new compounds were isolated through such a one-step reaction, and their structures were elucidated using various 2D-NMR techniques. This method provides easy access to multiple oxidative β-elemene derivatives in one single step and represents the first modifications on cyclohexyl ring of β-elemene. It is expected to open up the opportunity for future derivatization on cyclohexyl ring of β-elemene. The new compounds obtained above showed better anti-proliferation activities than β-elemene itself on several cancer cell lines. Among them, compound **17** shows the best activity in antiproliferation assays of A549 and U-87MG cell lines.

## 1. Introduction

The plant *Curcuma wenyujin* Y. H. Chen et C. Ling belongs to one of the important traditional Chinese medicines (TCM), and has been used to treat cancer and various diseases for nearly a thousand years [1–4]. The essential oil obtained from this plant is called elemene extracts, which contain at least four sesquiterpene isomers, namely α-elemene, β-elemene (**1**),

**Figure 1.** (*a*) The structure of β-elemene with carbon atoms numbering; (*b*) the ground-state chair conformation of β-elemene [18]. The two hydrogen atoms on C-2 and C-4 are theoretically accessible to SeO$_2$-mediated allylic oxidation besides C-13 and C-14.

γ-elemene, and δ-elemene [5–7]. In 2008, the State Food and Drug Administration of China approved the uses of elemene extracts in two special dose forms: liposomal oral liquids (for treatment of oesophageal and gastric cancers) and liposomal injections (to treat leukemia, brain, breast, ovarian and lung cancers) [8,9]. In the past two decades, numerous papers and patents published in China and across the world established the clinical usefulness of elemene extracts as a wide spectrum anti-cancer drug [10–16]. The mechanism of action of elemene is yet to be uncovered.

Within the four major sesquiterpene isomers, β-elemene is reported to be the major isomer, consisting of 40–80% of elemene extracts depending on the isolation and purification process [17]. It is undoubted that β-elemene is the major pharmacology contributor among the four isomers. Other isomers (α-elemene, γ-elemene and δ-elemene) might also contribute to the anti-cancer effects to some extent, which is another topic of interest under our investigation.

β-Elemene, named (5S,7R,10S)-(-)-(1-methyl-1-vinyl-2,4-diisopropenyl-cyclohexane), contains only hydrogen and carbon elements (figure 1). The biological activity of β-elemene is moderate or weak, indicated by its high IC$_{50}$ value against several tumour cell lines [19,20]. Barrero *et al.* reported the synthesis of β-elemene from germacrone in several steps [21]. In recent years, several papers have been published regarding the modifications of β-elemene, in the hope to seek better biological activity and to improve its water solubility [22–26]. The modifications of β-elemene described thus far are limited to two positions: C-13 and C-14. Such limitations are largely due to the stereoelectronic preference for reaction at C-13 and C-14. Herein, we report that SeO$_2$-mediated oxidation conditions can yield other oxidation patterns.

β-Elemene possesses three carbon-carbon double bonds, all connected to the cyclohexyl skeleton. These three C=C bonds are all terminal double bonds, two of them being di-substituted and one being mono-substituted. Conformational analysis of β-elemene suggests these three alkene substituents are probably positioned equatorial in ground state. Furthermore, C-14 is more sterically hindered than C-13 due to its proximity to the C-15 quaternary centre. Similarly, the proton on C-2 is slightly more hindered than the proton on C-4. The steric effects analysed above agree well with what Thomas *et al.* described in their epoxidation of β-elemene double bond [27].

Selenium dioxide (SeO$_2$)-mediated allylic oxidation of olefin to allylic alcohol, commonly known as Riley oxidation, is one of the most important transformations in organic synthesis [28,29]. Typically, an olefin is subjected to a catalytic SeO$_2$ and stoichiometric tert-butylhydroperoxide (TBHP) under mild conditions. Since its discovery, the Riley oxidation has been widely applied in organic synthesis [30,31]. The mechanism of Riley oxidation and the preferences and selectivity of reaction sites of the allylic group were well documented in the literature [32,33]. Preference (region- and chemoselectivity) will be dictated by stereoelectronics. In the case of β-elemene, there are four different allylic protons, namely protons at C-2, C-4, C-13 and C-14, respectively. Our interests in modifying unexplored positions of β-elemene prompt us to examine the SeO$_2$-TBHP condition on this substrate. Of all the four hydrogen-bearing allylic carbons, C-2 and C-4 draw our attention. We envision that the SeO$_2$-mediated oxidation reaction might access the hydrogen atoms on these two carbons, in addition to C-13 and C-14, and hence, might install the hydroxyl group on these two positions of cyclohexyl ring. As a result, the modification products of β-elemene on its cyclohexyl ring could be obtained. Those modifications on the cyclohexyl ring of β-elemene represent the synthetic challenge to date. Additionally, SeO$_2$-mediated allylic oxidation will also generate the oxidative products from C-13 and C-14 (plus the possible combination). These products, though previously reported, can only be synthesized in several steps [24,25,34–37] (figure 2).

**Figure 2.** Summary of the allylic oxidation of β-elemene.

## 2. Results

β-Elemene raw material is the gift from Holley Kingkong Pharmaceutical Co., Ltd. GC-MS analysis[1] suggests that it contains only about 78% of β-isomer, plus the other three isomers. Since the four elemene isomers possess very similar structure and physical properties, to purify them in the laboratory represents a challenge. Therefore, the material was used as-is. Understandably, the presence of other isomers will generally give lower yield as well as complicate the isolation process.

β-Elemene raw material was subjected to a SeO$_2$-mediated oxidation reaction in CH$_2$Cl$_2$, with 5 equivalent of TBHP at 0°C for 6 h. After standard work-up process, the crude product was purified in silica gel chromatography (petroleum ether (PE)/ethyl acetate (EA)) to yield four fractions, with polarity from the least to the most: fraction **I** ($R_f = 0.9$, PE/EA = 4:1, 7.4% yield), fraction **II** ($R_f = 0.7$, PE/EA = 4:1, 2.8% yield), fraction **III** ($R_f = 0.2$, PE/EA = 4:1, 8.6% yield) and fraction **IV** ($R_f = 0.15$, PE/EA = 4:1, 21.6% yield) (Scheme 1).

The above four fractions were analysed by HPLC. The results revealed that only fraction **IV** contains a single compound, the other three fractions were all mixtures of two compounds.

Fraction **I** appears to be a single spot in TLC (petroleum-ethyl acetate system). After screening with several mix solvent systems for TLC, petroleum ether/acetone system was found to be the best solvent to resolve the two compounds (Scheme 2). The structure of compound **2** was established by NMR in comparison with references [35]. The structure of compound **3** was elucidated through various 2D NMR techniques (see electronic supplementary material for details). It should be noted that compound **2** was synthesized in the literature involving three steps and a tedious HPLC purification process [35].

The $^1$H NMR of fraction **II** reveals two sets of signals in about 1 to 1 ratio, containing both aldehyde proton and allylic protons connecting to a hydroxyl group, presumably from compounds **6** and **7**. Attempt to resolve them in TLC using a variety of mix solvent systems (similar to fraction **I**) was proven to be unsuccessful. We then turned to protecting group strategy. Thus, 50 mg of fraction **II** was treated with TBDMS-Cl/imidazole to install TBDMS group on hydroxyl groups, resulting in two compounds **4** and **5**, which were carefully separated in silica gel chromatography. The isolation yield

---

[1]GC-MS analysis was performed in Agilent Technologies 7890B (GC system) and 5977A MSD (Mass unit) under the following conditions: Agilent gas chromatograph and gas work station, FID detector, capillary chromatography (Agilent 19091S-433UI, HP-5 ms Ultra Inert, 60–325°C, 30 m × 250 mm × 0.25 mm); injection temperature: 250°C, detector temperature: 230°C, rise range: starting temperature: 50°C, maintain for 2 min, rise to 80°C at the rate of 20°C per minute, maintained for 2 min, then increased to 150°C at the rate of 30°C per minute, maintained for 5 min; carrier gas: helium, flow rate: 24.2 ml min$^{-1}$, chromatographic column flow rate: 1.2 ml min$^{-1}$, pressure: 9.8 psi, tail gas flow rate: 3 ml min$^{-1}$, injection volume: 1 ml, split ratio: 100 : 1. The retention time of β-elemene is 9.05 min under the above conditions.

**Scheme 1.** Synthesis of β-elemene derivatives via SeO$_2$-mediated oxidation reaction.

**Scheme 2.** Compounds **2** and **3** were resolved in silica gel chromatography using petroleum ether/acetone mixed solvent.

is low because these two compounds are close to each other and the majority material stays as a mixture. If necessary, the mixture can be subjected to chromatography repeatedly to give a better yield of pure **4** and **5**. After standard deprotection of tert-butyldimethylsilyl group, compounds **6** and **7** were obtained respectively. The structures of **6** and **7** were established by NMR. Alternatively, compounds **6** and **7** might be separated using a suitable column in the preparative HPLC system (Scheme 3).

The $^1$H NMR of fraction **III** did not show the aldehyde proton signal. Therefore, the mixture presumably contains allylic alcohols. After screening a set of mixed solvent systems for TLC, dichloromethane/acetone (1:1 v/v) appears to give the best resolution. Thus compounds **8** and **9** were obtained in about 3 to 2 ratio, and their structures were established via various 2D NMR techniques (Scheme 4) [36].

The analysis by HPLC and $^1$H NMR of fraction **IV** indicates it is a single compound, whose structure was established as compound **10** by comparing its $^1$H NMR with known compound 13,14-bis(hydroxyl)-β-elemene [9,38,39]. Compound **10** is a known compound reported in the literature. It required a three-step synthesis using the literature method (reaction sequence: bis-allylic chlorination, nucleophilic displacement with OAc$^-$ and hydrolysis of ester), with a total yield of 19% [38]. Our method provides alternative access to this key intermediate in a single step.

To further expand the usefulness of SeO$_2$-mediated allylic oxidation reaction in β-elemene analogue synthesis, compound **12**[2] was subjected to a milder condition to see if we can obtain a higher yield and better regioselectivity of the oxidation product. To our delight, with less catalyst and oxidant (0.4 equivalent of SeO$_2$ and 0.8 equivalence of TBHP), the selective installation of the hydroxyl group was achieved in moderate to good yield. Compound **13** can serve as a key intermediate for further functionalize the C-13 and C-14 position of β-elemene (Scheme 5).

Huang reported that β-elemenal (**11**) (figure 3) showed better anti-proliferation activity against several tumour cell lines than β-elemene (**1**) itself [13]. Apparently the aldehyde functional group contributes to the biological activity. Thus, compounds **2**, **8**, **9** and **10** were converted to the

---

[2]β-Elemene was subjected to allylic chlorination in NaOCl/acetic acid to obtain a mixture of compound **11** and its regioisomer (i.e. Cl-group in 14-position of β-elemene). After repeated chromatography in preparative HPLC, compound **11** was obtained in pure form.

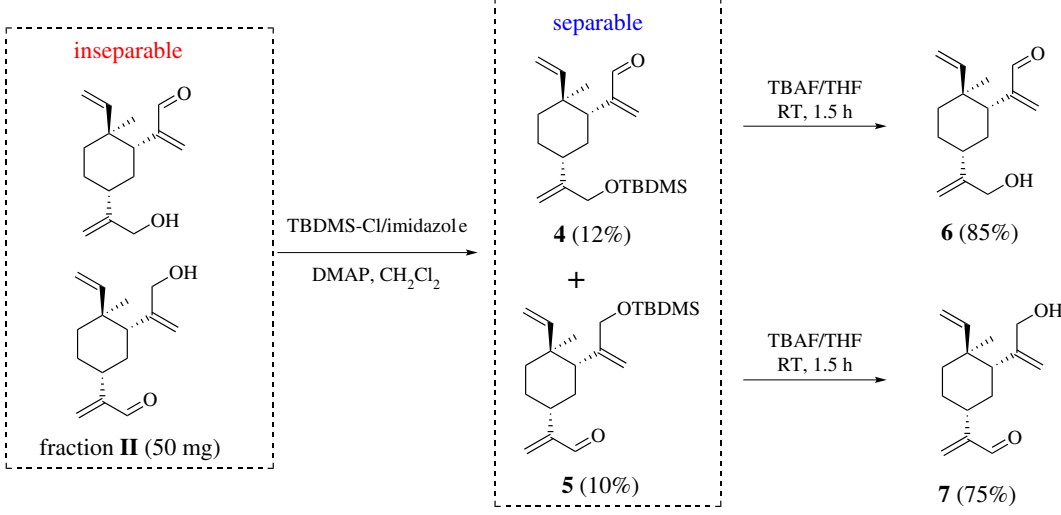

**Scheme 3.** Compounds **6** and **7** were resolved through protecting group installation and deprotection process.

**Scheme 4.** Compounds **8** and **9** were resolved in silica gel chromatography using dichloromethane/acetone mixed solvent.

**Scheme 5.** Synthesis of compound **13** via SeO₂-mediated oxidation reaction.

**Figure 3.** Structure of compound **10** and elemenal (**11**).

corresponding aldehydes **14–17** using standard oxidation conditions (pyridinium dichromate (PDC) in dichloromethane, 0°C to RT) (Scheme 6). All compounds were characterized by $^1$H NMR. Undoubtedly, compounds **14–17** can serve as key intermediates in further functionalization of β-elemene.

All new compounds and β-elemene were subjected to cell proliferation inhibition assay against two tumour cell lines: A549 and U-87MG. The results will be discussed below.

**Scheme 6.** Synthesis of β-elemene derivatives via PDC-mediated oxidation reaction.

**Table 1.** Inhibition of cell proliferation against A549 and U-87MG cell lines.[a]

| Compound | A549 (IC$_{50}$, μM)[b] | U-87 (IC$_{50}$, μM)[b] |
|---|---|---|
| β-elemene (**1**)[c] | >300 | >300 |
| **2** | >100 | >100 |
| **3** | >100 | >100 |
| **6** | 87.23 ± 0.2 | 32.92 ± 0.3 |
| **7** | 44.57 ± 0.1 | 14.31 ± 0.4 |
| **8** | >100 | >100 |
| **9** | >100 | >100 |
| **10** | >100 | >100 |
| **13** | 41.24 ± 0.5 | 40.35 ± 0.2 |
| **14** | >100 | >100 |
| **15** | 45.28 ± 0.3 | 25.45 ± 0.1 |
| **16** | 91.44 ± 0.2 | >100 |
| **17** | 9.34 ± 0.1 | 2.83 ± 0.4 |
| **STS**[d] | 0.021 ± 0.003 | 0.230 ± 0.01 |

[a]IC$_{50}$ (μM): inhibitory concentration of 50% cell growth was calculated through a nonlinear fit-curve (log of compound concentration versus normalized response—variable slope).
[b]Data are presented as the means ± s.d. of three independent experiments. $p < 0.05$.
[c]The material tested here contains 78% of β-elemene, which is the same batch used in the experimental section.
[d]STS (staurosporine) was used as the positive control.

The β-elemene derivatives obtained above were assessed in anti-proliferation assay against A549 and U-87MG cell lines. The activity of some representative compounds is shown in table 1. Compounds **6**, **7**, **11**, **15** and **17** showed improved inhibitory activities than β-elemene. It is interesting to note that all the compounds showing better biological activities possess an aldehyde functional group. The reason for such phenomenon is currently being investigated in our laboratory.

## 3. Experimental

Typical procedure: to a solution of β-elemene (**1**) (500 mg, 2.45 mmol, 78%) in CH$_2$Cl$_2$ (10 ml) at 0°C was added dropwise 65% aqueous tert-butyl hydroperoxide (1.7 ml, 12.21 mmol). SeO$_2$ (270 mg, 2.45 mmol) was added into the above mixture. Then the mixture was stirred at 0°C for 6 h. The reaction was

quenched at 0°C with saturated aqueous NaHSO$_3$ solution (15 ml). The organic layer was separated, the aqueous layer was extracted with CH$_2$Cl$_2$ (2 × 10 ml). The combined organic extracts were washed with brine (10 ml), dried over anhydrous Na$_2$SO$_4$ and concentrated under reduced pressure. The residue was purified via silica gel column chromatography (petroleum ether/ethyl acetate) to yield four fractions, with polarity from the least to the most: fraction **I** (**2** and **3**, 40 mg, 7.4%, R$_f$ = 0.9, PE/EA = 4:1), fraction **II** (**6** and **7**, 16 mg, 2.8%, R$_f$ = 0.7, PE/EA = 4:1), fraction **III** (**8** and **9**, 50 mg, 8.6%, R$_f$ = 0.2, PE/EA = 4:1) and fraction **IV** (**10**, 125 mg, yield 21.6%, R$_f$ = 0.15, PE/EA = 4:1). Fraction **I** was further purified in silica gel chromatography (petroleum ether/acetone, v/v = 10:1) to yield compounds **2** (27 mg, yield 5.0%) and **3** (9 mg, yield 1.7%). Fraction **III** was further purified in silica gel chromatography (dichloromethane/acetone, v/v = 10:1) to yield compounds **8** (27 mg, yield 4.7%) and **9** (18 mg, yield 3.1%).

**2-((1R,2S,5R)-2-methyl-5-(prop-1-en-2-yl)-2-vinylcyclohexyl)prop-2-en-1-ol (2)** [35]: colourless oil. $^1$H NMR (400 MHz, CDCl$_3$) δ 5.77 (dd, J = 17.8, 10.5 Hz, 1H), 5.17 (q, J = 1.5 Hz, 1H), 4.96–4.92 (m, 1H), 4.90 (q, J = 1.3 Hz, 1H), 4.85–4.83 (m, 1H), 4.72 (m, 2H), 4.09–3.94 (m, 2H), 2.07–2.00 (m, 1H), 2.00–1.90 (m, 1H), 1.75 (t, J = 1.1 Hz, 3H), 1.66–1.43 (m, 6H), 1.00 (s, 3H). HRMS (ESI) calcd for C$_{15}$H$_{24}$NaO [M + Na]$^+$: 243.1719, found 243.1725.

**(1S,3S,4S)-4-methyl-1,3-di(prop-1-en-2-yl)-4-vinylcyclohexan-1-ol (3)**: colourless oil. $^1$H NMR (400 MHz, CDCl$_3$) δ 5.75 (dd, J = 17.8, 10.5 Hz, 1H), 5.06 (m, 2H), 4.93–4.90 (m, 1H), 4.88 (s, 1H), 4.83 (p, J = 1.6 Hz, 1H), 4.61 (dt, J = 1.9, 0.9 Hz, 1H), 2.06–1.85 (m, 3H), 1.81 (t, J = 1.0 Hz, 3H), 1.70 (dd, J = 1.5, 0.8 Hz, 3H), 1.67–1.28 (m, 4H), 1.07 (s, 3H). $^{13}$C NMR (126 MHz, CDCl$_3$) δ 149.69, 146.62, 146.47, 113.35, 112.60, 110.36, 74.39, 49.64, 39.72, 37.47, 37.20, 31.77, 24.39, 18.71, 16.68. HRMS (ESI) calcd for C$_{15}$H$_{24}$NaO [M + Na]$^+$: 243.1719, found 243.1722.

**(1S,2S,5R)-1-(3-hydroxyprop-1-en-2-yl)-2-methyl-5-(prop-1-en-2-yl)-2-vinylcyclohexanol (8)**: colourless oil. $^1$H NMR (400 MHz, CDCl$_3$) δ 5.69 (dd, J = 17.7, 10.5 Hz, 1H), 5.19 (q, J = 1.4 Hz, 1H), 5.07 (m, 2H), 4.97–4.91 (m, 1H), 4.89 (d, J = 6.9 Hz, 2H), 4.09–3.86 (m, 2H), 2.08–1.99 (m, 2H), 1.98–1.85 (m, 2H), 1.81 (d, J = 1.2 Hz, 3H), 1.72 (td, J = 13.8, 3.9 Hz, 1H), 1.47 (td, J = 13.9, 3.6 Hz, 1H), 1.34 (dt, J = 13.7, 3.7 Hz, 1H), 1.06 (s, 3H). $^{13}$C NMR (126 MHz, CDCl$_3$) δ 150.53, 149.08, 146.29, 113.60, 111.67, 111.22, 74.32, 67.42, 44.90, 39.60, 37.92, 37.06, 31.60, 18.74, 16.04. HRMS (ESI) calcd for C$_{15}$H$_{24}$O$_2$ [M + H]$^+$: 236.1776, found 236.1771.

**(1S,3R,4S)-3-(3-hydroxyprop-1-en-2-yl)-4-methyl-1-(prop-1-en-2-yl)-4-vinylcyclohexanol (9)**: colourless oil. $^1$H NMR (400 MHz, CDCl$_3$) δ 5.82 (dd, J = 17.8, 10.5 Hz, 1H), 5.20 (d, J = 1.5 Hz, 1H), 5.05 (s, 1H), 4.97–4.94 (m, 1H), 4.94–4.90 (m, 1H), 4.83 (d, J = 1.4 Hz, 2H), 4.10–3.95 (m, 2H), 2.49 (dd, J = 13.3, 3.3 Hz, 1H), 1.94 (m, 2H), 1.90–1.85 (m, 1H), 1.84 (s, 3H), 1.49 (ddt, J = 16.7, 13.9, 2.9 Hz, 2H), 1.31–1.28 (m, 1H), 0.99 (s, 3H). $^{13}$C NMR (126 MHz, CDCl$_3$) δ 152.04, 150.87, 149.20, 111.47, 110.91, 109.25, 73.99, 67.76, 42.14, 39.45, 37.69, 34.66, 31.02, 19.00, 15.07. HRMS (ESI) calcd for C$_{15}$H$_{24}$O$_2$ [M + H]$^+$: 236.1776, found 236.1779.

**2,2′-((1R,3R,4S)-4-methyl-4-vinylcyclohexane-1,3-diyl)bis(prop-2-en-1-ol) (10)** [36]: colourless oil. $^1$H NMR (400 MHz, CDCl$_3$) δ 5.77 (dd, J = 17.8, 10.5 Hz, 1H), 5.17 (q, J = 1.4 Hz, 1H), 5.06 (d, J = 1.6 Hz, 1H), 4.94 (m, 2H), 4.90 (q, J = 1.4 Hz, 1H), 4.84 (s, 1H), 4.14 (s, 2H), 4.10–3.93 (m, 2H), 2.11–1.99 (m, 2H), 1.70–1.28 (m, 6H), 1.02 (s, 3H). HRMS (ESI) calcd for C$_{15}$H$_{24}$O$_2$ [M + H]$^+$: 236.1776, found 236.1772.

To a solution of fraction **II** (122 mg, 0.52 mmol), imidazole (53 mg, 0.78 mmol) and 4-dimethylaminopyridine (3 mg, 0.025 mmol) in CH$_2$Cl$_2$ (5 ml) at room temperature was added tert-butyldimethylsilyl chloride (118 mg, 0.78 mmol). The mixture was stirred at this temperature for 8 h. The reaction solution was diluted with H$_2$O (2 ml) and extracted with ethyl acetate (3 × 6 ml). The combined organic layers were washed with brine (4 ml), dried over Na$_2$SO$_4$. After filtering, the filtrate was concentrated under reduced pressure. The residue was purified via column chromatography to afford the compounds **4** (22 mg, yield 12%) and **5** (18 mg, yield: 10%).

To a solution of **5** (18 mg, 0.052 mmol) in dry THF (2 ml) was added tetrabutylammonium fluoride (0.52 ml, 0.52 mmol, 1 M solution in THF), and mixture was stirred at room temperature for 1.5 h. The reaction solution was diluted with H$_2$O (1 ml). The THF was evaporated under reduced pressure and the residue was extracted with ethyl acetate (3 × 5 ml). The combined organic layers were washed with saturated brine (3 ml), dried over Na$_2$SO$_4$. After filtering, the filtrate was concentrated under reduced pressure. The residue was purified via column chromatography to afford the compound **7** (9 mg, yield 75%). $^1$H NMR (500 MHz, CDCl$_3$) δ 9.52 (s, 1H), 6.29 (d, J = 1.1 Hz, 1H), 5.99 (s, 1H), 5.78 (dd, J = 17.8, 10.5 Hz, 1H), 5.17 (d, J = 1.4 Hz, 1H), 4.97–4.88 (m, 2H), 4.82 (s, 1H), 4.11–3.91 (m, 2H), 2.56 (dq, J = 11.8, 7.2, 5.8 Hz, 1H), 2.13–2.07 (m, 1H), 1.67–1.61 (m, 3H), 1.61–1.55 (m, 2H), 1.53–1.45 (m, 1H), 1.03 (s, 3H). $^{13}$C NMR (126 MHz, CDCl$_3$) δ 194.51, 154.57, 151.12, 149.31, 132.99, 111.01, 110.95, 67.37, 47.82, 39.58, 39.49, 36.56, 33.17, 26.61, 16.04. HRMS (ESI) calcd for C$_{15}$H$_{22}$O$_2$ [M + H]$^+$: 234.1620, found 234.1628.

Using the procedure above, compound **6** (12 mg, yield 85%) was prepared from compound **4** (22 mg, 0.062 mmol). $^1$H NMR (500 MHz, CDCl$_3$) $\delta$ 9.39 (s, 1H), 6.15 (s, 1H), 6.04 (s, 1H), 5.65 (dd, $J = 17.4$, 10.8 Hz, 1H), 5.07 (d, $J = 1.4$ Hz, 1H), 4.95 (t, $J = 1.2$ Hz, 1H), 4.85–4.70 (m, 2H), 4.13 (d, $J = 1.2$ Hz, 2H), 2.89 (dd, $J = 13.1$, 3.3 Hz, 1H), 2.18–2.08 (m, 1H), 1.73–1.66 (m, 1H), 1.66–1.55 (m, 2H), 1.54–1.49 (m, 2H), 1.28 (m, 1H), 0.95 (s, 3H). $^{13}$C NMR (126 MHz, CDCl$_3$) $\delta$ 194.40, 153.34, 151.94, 148.78, 135.12, 110.69, 108.27, 65.21, 41.31, 41.29, 39.53, 39.40, 32.60, 27.12, 14.93. HRMS (ESI) calcd for C$_{15}$H$_{22}$O$_2$ [M + H]$^+$: 234.1620, found 234.1627.

To a stirred solution of compound **2** (50 mg, 0.23 mmol) in CH$_2$Cl$_2$ (5 ml) was added PDC (129 mg, 0.34 mmol) at 0°C. The solution was sealed and stirred at 0°C for 1 h, then at room temperature for 12 h. The precipitate was filtered and washed with CH$_2$Cl$_2$. The filtrate was evaporated under reduced pressure. The residue was purified via column chromatography to afford compound **14** [37] (20 mg, yield 40%). $^1$H NMR (400 MHz, CDCl$_3$) $\delta$ 9.32 (s, 1H), 6.08 (s, 1H), 5.97 (s, 1H), 5.59 (dd, $J = 17.4$, 10.8 Hz, 1H), 4.70 (dd, $J = 36$, 1.4 Hz, 1H), 4.71 (t, $J = 1.6$ Hz, 1H), 4.65–4.63 (m, 2H), 2.82 (dd, $J = 13.0$, 3.4 Hz, 1H), 2.10–0.93 (m, 7H), 1.67 (d, $J = 1.2$ Hz, 3H), 0.87 (s, 3H). $^{13}$C NMR (126 MHz, CDCl$_3$) $\delta$ 194.48, 152.15, 149.91, 148.97, 135.06, 110.56, 108.51, 45.35, 41.20, 39.55, 39.37, 32.16, 26.65, 21.10, 14.92. HRMS (ESI) calcd for C$_{15}$H$_{22}$O [M + H]$^+$: 218.1671, found 218.1676.

Compounds **15**, **16**, **17** were prepared by following the same procedure as those described for **14**.

**2-((1R,2S,5S)-5-hydroxy-2-methyl-5-(prop-1-en-2-yl)-2-vinylcyclohexyl)acrylaldehyde 15** (yield 38.7%): colourless oil. $^1$H NMR (400 MHz, CDCl$_3$) $\delta$ 9.40 (s, 1H), 6.13 (s, 1H), 6.06 (s, 1H), 5.72 (dd, $J = 17.5$, 10.8 Hz, 1H), 5.05 (dd, $J = 1.5$, 0.8 Hz, 1H), 4.85–4.75 (m, 3H), 3.36 (dd, $J = 13.6$, 3.4 Hz, 1H), 2.02 (td, $J = 13.6$, 4.9 Hz, 2H), 1.90–1.82 (m, 1H), 1.85 (dd, $J = 1.4$, 0.7 Hz, 3H), 1.59–1.51 (m, 1H), 1.41–1.27 (m, 3H), 0.93 (s, 3H). $^{13}$C NMR (126 MHz, CDCl$_3$) $\delta$ 194.88, 165.86, 145.34, 142.32, 127.42, 110.80, 109.34, 83.42, 39.51, 35.90, 31.88, 29.70, 29.49, 29.14, 14.15. HRMS (ESI) calcd for C$_{15}$H$_{22}$O$_2$ [M + H]$^+$: 234.1620, found 234.1624.

**2-((1R,2S,5R)-5-hydroxy-2-methyl-5-(prop-1-en-2-yl)-2-vinylcyclohexyl)acrylaldehyde 16** (yield 36%): colourless oil. $^1$H NMR (400 MHz, CDCl$_3$) $\delta$ 9.29 (s, 1H), 7.54 (s, 1H), 5.88 (ddd, $J = 18.0$, 10.7, 1.0 Hz, 1H), 5.15 (d, $J = 1.9$ Hz, 1H), 5.13–5.10 (m, 1H), 5.00 (s, 1H), 4.90 (t, $J = 1.5$ Hz, 1H), 2.87 (s, 1H), 2.09 (dd, $J = 13.3$, 2.7 Hz, 1H), 1.88–1.81 (m, 3H), 1.80 (t, $J = 1.0$ Hz, 3H), 1.61 (dt, $J = 3.9$, 1.9 Hz, 1H), 1.46 (ddd, $J = 13.4$, 3.8, 1.9 Hz, 1H), 1.40–1.32 (m, 1H), 0.85 (s, 3H). $^{13}$C NMR (126 MHz, CDCl$_3$) $\delta$ 189.26, 167.08, 146.91, 144.95, 121.31, 112.93, 111.21, 83.42, 40.80, 33.33, 32.69, 29.86, 29.70, 28.54, 18.63. HRMS (ESI) calcd for C$_{15}$H$_{22}$O$_2$ [M + H]$^+$: 234.1620, found 234.1627.

**2,2′-((1R,3R,4S)-4-methyl-4-vinylcyclohexane-1,3-diyl)diacrylaldehyde 17** (yield 80%): colourless oil. $^1$H NMR (400 MHz, CDCl$_3$) $\delta$ 9.53 (s, 1H), 9.38 (s, 1H), 6.29 (d, $J = 1.0$ Hz, 1H), 6.12 (s, 1H), 6.03 (s, 1H), 6.00 (s, 1H), 5.67 (dd, $J = 17.4$, 10.8 Hz, 1H), 4.84–4.73 (m, 2H), 2.95 (dd, $J = 13.1$, 3.3 Hz, 1H), 2.67–2.57 (m, 1H), 1.73–1.21 (m, 6H), 0.97 (s, 3H). $^{13}$C NMR (126 MHz, CDCl$_3$) $\delta$ 194.36, 194.21, 154.35, 151.71, 148.65, 135.04, 133.11, 110.78, 41.35, 39.17, 36.77, 31.73, 29.70, 26.46, 18.45. HRMS (ESI) calcd for C$_{15}$H$_{20}$O$_2$ [M + H]$^+$: 232.1463, found 232.1469.

To a solution of **12** (100 mg, 0.42 mmol) in CH$_2$Cl$_2$ (3 ml) at 0°C was added dropwise 65% aqueous tert-butyl hydroperoxide (0.047 ml, 0.336 mmol). SeO$_2$ (19 mg, 0.168 mmol) was added into the above mixture. The mixture was stirred at 0°C for 6 h. The reaction was quenched at 0°C with saturated aqueous NaHSO$_3$ solution (3 ml). The organic layer was separated, the aqueous layer was extracted with CH$_2$Cl$_2$ (2 × 3 ml). The combined organic extracts were washed with brine (3 ml), dried over anhydrous Na$_2$SO$_4$ and concentrated under reduced pressure. The residue was purified via silica gel column chromatography (petroleum ether/ethyl acetate) to afford compound **13** (60 mg, yield 56%). $^1$H NMR (500 MHz, CDCl$_3$) $\delta$ 5.77 (dd, $J = 17.7$, 10.5 Hz, 1H), 5.21–5.14 (m, 2H), 5.04 (d, $J = 1.1$ Hz, 1H), 4.97–4.90 (m, 2H), 4.84 (s, 1H), 4.09 (d, $J = 0.9$ Hz, 2H), 4.09–3.94 (m, 2H), 2.28–2.17 (m, 1H), 2.07 (dd, $J = 12.3$, 3.9 Hz, 1H), 1.77–1.69 (m, 1H), 1.69–1.57 (m, 1H), 1.57–1.44 (m, 4H), 1.01 (s, 3H). $^{13}$C NMR (126 MHz, CDCl$_3$) $\delta$ 151.20, 149.59, 149.35, 113.27, 111.22, 110.94, 67.48, 47.90, 47.69, 41.07, 39.65, 39.60, 33.64, 27.00, 16.01. HRMS (ESI) calcd for C$_{15}$H$_{23}$ClO [M + H]$^+$: 254.1437, found 254.1439.

# 4. Biological assay

## 4.1. Materials

Cell Titer-Glo luminescent cell viability assay kits (cat. no. G7573, lot. no. 0000365004) were obtained from Promega; MEM (cat. no. 11095-080; lot. no. 2053121), 0.25% Trypsine-EDTA (cat. no. 25200-072, lot. no. 2042303) and F-12 K (cat. no. 21127-022; lot. no. 2085296) were obtained from Invitrogen; FBS

(Biological Industries, cat. no. 04-002-1A, lot. no. 1841929); Penicillin-Streptomycin solution cat. no. SV30010, lot. no. J180029) was obtained from Hyclone; Glutamax (cat. no. 35050-061; lot no. 2085268) was obtained from Gibco; DMSO (cat. no. 276855-1 L, lot. no. STBD7938 V) was obtained from Sigma; 96-well plate, white wall with clear bottom, tissue culture-treated (cat. no. CLS3903; lot. no. 25116010) was obtained from Corning.

## 4.2. Methods

The suspension of specific tumour cells was adjusted to $5 \times 10^4$ ml$^{-1}$ or $2 \times 10^4$ ml$^{-1}$ with DMEM + 2 mm glutamine + 10% FBS medium. Add 100 µl cell suspension to 96-well cell culture plate, and the final cell concentration is 5000 cells well$^{-1}$ (72 h). DMSO was used to dissolve the compound to be tested as 100 mM storage solution. The final concentration of 200× compound was prepared with storage solution and DMSO, and the gradient dilutions of 3× series were prepared, and then diluted 20 times with culture medium respectively. Finally, 10 µl corresponding 10-fold solution was added to each cell hole, and each drug concentration was in duplicate holes. The final concentrations of each compound were 300 µM, 100 µM, 33.33 µM, 11.11 µM, 3.704 µM, 1.235 µM and 0.412 µM, and the final concentration of DMSO per pore was 0.5%. Incubate in a 37°C, 5% $CO_2$ incubator for 72 h. After 72 h of drug treatment, add 100 µl celltiter glo detection reagent into each hole according to the CTG operation instructions, melt and balance the CTG solution to room temperature in advance, mix it with microporous plate shaker for 2 min, place it at room temperature for 10 min, and the luminescence is recorded with a luminometer. The cell survival rate was calculated by the formula: $(V_{\text{sample}} - V_{\text{blank}})/(V_{\text{vehicle control}} - V_{\text{blank}}) \times 100\%$. $V_{\text{sample}}$ is the reading of the drug treatment group, $V_{\text{vehicle control}}$ is the average of the solvent control group, and $V_{\text{blank}}$ is the average of the blank control hole. By using graphpad prism 5.0 software, a nonlinear regression model was used to draw the S-type dose survival curve and calculate the IC$_{50}$ value.

# 5. Conclusion

SeO$_2$-mediated allylic oxidation reaction was first applied to β-elemene, a substrate bearing three carbon-carbon double bonds and several allylic hydrogen atoms, and was discovered to produce seven derivatives of β-elemene in a single step. Several additional analogues of β-elemene were further synthesized and found to display better inhibitory activities against A549 and U-87 cell proliferation. All compounds in this article can serve as key intermediates for further derivatization of β-elemene. Our approaches represent the first success to introduce functional groups onto the cyclohexyl ring, a challenging task unexplored before.

Data accessibility. Supporting information includes $^1$H NMR, $^{13}$C NMR, $^1$H-$^1$H COSY, HMBC, HSQC, NOESY spectra of compounds **3**, **8** and **9**. Our data are deposited at the Dryad Digital Repository: https://doi.org/10.5061/dryad.cnp5hqc1w [40].

Authors' contributions. X.H. and X.-T.Z. conducted the synthesis of all the compounds in this manuscript equally. Y.G. analysed NMR raw data and elucidated the structures for compounds **3**, **8** and **9**. R.B. performed the anti-proliferation assay in tumour cell lines. X.-Y.Y. and T.X. conceived the idea, designed the project and supervised the research work. They also completed the writing of entire manuscript together.

Competing interests. The authors declare no competing interests.

Funding. This work was funded by the National Natural Science Foundation of China (81730108 and 81973635), Hangzhou Normal University startup fund (4125C5021920419), the Natural Science Foundation of Shandong Province (ZR2019BB046) and the open project of Shandong Collaborative Innovation Center for Antibody Drugs (CIC-AD1823).

Acknowledgements. The authors thank the following individuals or organizations: Dalian Holley Kingkong Pharmaceutical Co., Ltd for providing β-elemene (78% purity) as a gift; Prof. Kefang Yang of Key Laboratory of Organosilicon Chemistry and Material Technology of Ministry of Education, Hangzhou Normal University for recording NMR data.

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
