## [Reviewer comments · Royal Society Open Science]

Review History

RSOS-200038.R0 (Original submission)

Review form: Reviewer 1

Is the manuscript scientifically sound in its present form?

No

Are the interpretations and conclusions justified by the results?

Yes

Is the language acceptable?

No

Do you have any ethical concerns with this paper?

No

Have you any concerns about statistical analyses in this paper?

No

Recommendation?

Major revision is needed (please make suggestions in comments)

Comments to the Author(s)

Xie and coworkers describe that beta-elemene can react with SeO₂ to yield previously inaccessible oxidation patterns on the core skeleton. The conditions are not selective, but nonetheless yield novel entities of interest to elemene-based research. Also described are conditions for purifying these novel oxidation products. Finally, the authors report an analysis of the antiproliferative activity of these molecules on several cancer cell lines.

Also presented is a background on elemene and summary of previous synthetic efforts that described allylic chlorination of elemene. On this line, a figure summarizing known semisynthesis and this report would be valuable.

- I have included a sample of grammatical corrections for two pages. The manuscript is very difficult to read. The document as a whole needs significant writing improvement (Appendix A).
- Journal abbreviations are often incorrect. For example, I don't know and could never find reference 9 (what journal is just "Chemistry?"). Ref. 20, 24, 27, 28, 29, 30, etc are all incorrect abbreviations. This made it so I couldn't easily copy and paste references to check for accuracy. I strongly recommend a more close analysis of references prior to submission as a courtesy to referees.
- I could see a summary of the known elemene semisynthesis being a valuable figure for the paper. Consider adding one.
- Elemene should not be capitalized when used in the middle of a sentence (do not capitalize names of natural products). This is inconsistent throughout the manuscript.
- Consider revising certain figures for effective use of "white space" and aesthetics.
- The NMRs look correct, though a few of the ¹³C NMRs could have better S/N ratios.

Review form: Reviewer 2 (Lingamallu Rao)

Is the manuscript scientifically sound in its present form?

Yes

Are the interpretations and conclusions justified by the results?

No

Is the language acceptable?

Yes

Do you have any ethical concerns with this paper?

No

Have you any concerns about statistical analyses in this paper?

Yes

Recommendation?

Accept with minor revision (please list in comments)

Comments to the Author(s)

Authors attempted to describe β -Elemene derivatives produced from SeO₂-mediated oxidation reaction.

The MS may be considered for publication after receiving proper responses for the following points:

1. What is the status of the prepared compounds, when compared to standard drugs used for inhibition of proliferation of the cancer cell lines.

Table 1. Include IC 50 values of selected standard drugs for Inhibition of cell proliferation against A549 and U-87MG cell lines for comparison.

2. Include the IC 50 values of other elemenes (alpha, gamma and delta) in table 1.

3. Authors may through light on the mechanism of improved action on the cell lines.

4. Whether highly active compound 17(in the MS) is new? It may be highlighted in the abstract.

5. Authors may include statistical analysis for the activity studies.

6. Conclusion-last sentences: Our approaches represent the first success to modify cyclohexyl ring, a challenging task unexplored before.

Cyclohexyl ring was not modified. Oxygen functional groups introduced on the cyclohexyl ring of beta elemene. Authors may modify appropriately

7. ¹H NMR and ¹³C NMR data for all the compounds may be presented in two tables (one for each ¹H and ¹³C) along with assignments (proton and carbon numbers)

8. The following references may be included at appropriate place in MS.

(i) Alan F. Thomas, Christian Vial, Michel Ozainne, Günter Ohloff; The Oxidation of the Double Bonds of β -Elemene; October 2004, Helvetica Chimica Acta 56(7):2270 – 2279, DOI:

10.1002/hlca.19730560715;

(ii) Jacek Młochowski and Halina Wójtowicz-Młochowska; Developments in Synthetic Application of Selenium(IV) Oxide and Organoselenium Compounds as Oxygen Donors and Oxygen-Transfer Agents; Molecules 2015, 20, 10205-10243;

Decision letter (RSOS-200038.R0)

28-Feb-2020

Dear Dr Ye:

Title: β -Elemene derivatives produced from SeO₂-mediated oxidation reaction

Manuscript ID: RSOS-200038

The editor assigned to your manuscript has now received comments from reviewers. We would like you to revise your paper in accordance with the referee and Subject Editor suggestions which can be found below (not including confidential reports to the Editor). Please note this decision does not guarantee eventual acceptance.

Please submit your revised paper before 22-Mar-2020. Please note that the revision deadline will expire at 00.00am on this date. If we do not hear from you within this time then it will be assumed that the paper has been withdrawn. In exceptional circumstances, extensions may be possible if agreed with the Editorial Office in advance. We do not allow multiple rounds of revision so we urge you to make every effort to fully address all of the comments at this stage. If deemed necessary by the Editors, your manuscript will be sent back to one or more of the original reviewers for assessment. If the original reviewers are not available we may invite new reviewers.

On behalf of the Subject Editor Professor Anthony Stace and the Associate Editor Dr Andrew Harned.

RSC Associate Editor:

Comments to the Author:

The referees have returned several comments and suggestions. The request for IC50 data using known antiproliferative agents is an important control in order to demonstrate the veracity of your assay. Although Reviewer 2 has requested tabulated NMR data, I do not feel this request adds to the manuscript and is not in line with the style of the journal. Please keep the data in its current location.

In addition to the comments raised by the reviewers, I have a couple of additional concerns that should be incorporated into a revised manuscript. I see that the supplied characterization data for several compounds is missing ¹³C NMR and HRMS data. If these compounds have been previously reported in the literature, please supply an appropriate reference along with your own measured ¹H NMR data for comparison. If these are new compounds, then full characterization must be provided. Also, in addition to the percent yield that is already present in the experimental procedures, please provide the actual yield as a mass.

RSC Subject Editor:

Comments to the Author:

(There are no comments.)

Reviewers' Comments to Author:

Reviewer: 1

Comments to the Author(s)

Xie and coworkers describe that beta-elemene can react with SeO₂ to yield previously inaccessible oxidation patterns on the core skeleton. The conditions are not selective, but nonetheless yield novel entities of interest to elemene-based research. Also described are

conditions for purifying these novel oxidation products. Finally, the authors report an analysis of the antiproliferative activity of these molecules on several cancer cell lines.

Also presented is a background on elemene and summary of previous synthetic efforts that described allylic chlorination of elemene. On this line, a figure summarizing known semisynthesis and this report would be valuable.

- I have included a sample of grammatical corrections for two pages. The manuscript is very difficult to read. The document as a whole needs significant writing improvement.
- Journal abbreviations are often incorrect. For example, I don't know and could never find reference 9 (what journal is just "Chemistry?"). Ref. 20, 24, 27, 28, 29, 30, etc are all incorrect abbreviations. This made it so I couldn't easily copy and paste references to check for accuracy. I strongly recommend a more close analysis of references prior to submission as a courtesy to referees.
- I could see a summary of the known elemene semisynthesis being a valuable figure for the paper. Consider adding one.
- Elemene should not be capitalized when used in the middle of a sentence (do not capitalize names of natural products). This is inconsistent throughout the manuscript.
- Consider revising certain figures for effective use of "white space" and aesthetics.
- The NMRs look correct, though a few of the ¹³C NMRs could have better S/N ratios.

Reviewer: 2

Comments to the Author(s)

Authors attempted to describe β -Elemene derivatives produced from SeO₂-mediated oxidation reaction.

The MS may be considered for publication after receiving proper responses for the following points:

1. What is the status of the prepared compounds, when compared to standard drugs used for inhibition of proliferation of the cancer cell lines.
Table 1. Include IC 50 values of selected standard drugs for Inhibition of cell proliferation against A549 and U-87MG cell lines for comparison.
2. Include the IC 50 values of other elemenes (alpha, gamma and delta) in table 1.
3. Authors may through light on the mechanism of improved action on the cell lines.
4. Whether highly active compound 17(in the MS) is new? It may be highlighted in the abstract.
5. Authors may include statistical analysis for the activity studies.
6. Conclusion-last sentences: Our approaches represent the first success to modify cyclohexyl ring, a challenging task unexplored before.
Cyclohexyl ring was not modified. Oxygen functional groups introduced on the cyclohexyl ring of beta elemene. Authors may modify appropriately
7. ¹H NMR and ¹³C NMR data for all the compounds may be presented in two tables (one for each ¹H and ¹³C) along with assignments (proton and carbon numbers)
8. The following references may be included at appropriate place in MS.
 - (i) Alan F. Thomas, Christian Vial, Michel Ozainne, Günter Ohloff; The Oxidation of the Double Bonds of β -Elemene; October 2004, Helvetica Chimica Acta 56(7):2270 – 2279, DOI: 10.1002/hlca.19730560715;
 - (ii) Jacek Młochowski and Halina Wójtowicz-Młochowska; Developments in Synthetic Application of Selenium(IV) Oxide and Organoselenium Compounds as Oxygen Donors and Oxygen-Transfer Agents; Molecules 2015, 20, 10205-

Author's Response to Decision Letter for (RSOS-200038.R0)

See Appendix B.

Decision letter (RSOS-200038.R1)

26-Mar-2020

Dear Dr Ye:

Title: β -Elemene derivatives produced from SeO₂-mediated oxidation reaction
Manuscript ID: RSOS-200038.R1

It is a pleasure to accept your manuscript in its current form for publication in Royal Society Open Science. The chemistry content of Royal Society Open Science is published in collaboration with the Royal Society of Chemistry.

On behalf of the Subject Editor Professor Anthony Stace and the Associate Editor Dr Andrew Harned.

RSC Associate Editor

Comments to the Author:

The authors appear to have addressed all of the concerns raised by the previous review (either in the main text or in their response letter). I can now recommend publication of the current manuscript.

Reviewer(s)' Comments to Author:

Appendix A

cell lines.^{18, 19} Barrero *et al* reported the synthesis of β -elemene from germacrone in several steps.²⁰ In recent years, several papers have been published regarding the modifications of β -elemene, in the hope to seek better biological activity and to improve its water solubility.²¹⁻²⁵ However, up to date, the modifications of β -elemene are limited ~~only~~ to two positions: C-13 and C-14. Such limitation is largely due to the difficulty to introduce functional groups to the positions other than C-13 and C-14. ~~up to date, the chemistries to modify those positions are unexplored.~~

Figure 1. (a) The structure of β -elemene with carbon atoms numbering; (b) The chair conformation of β -elemene.²⁶ The two hydrogen atoms on C-2 and C-4 are theoretically accessible to SeO_2 -mediated allylic oxidation besides C-13 and C-14.

β -Elemene possesses three carbon-carbon double bonds, all connected to the cyclohexyl skeleton. These three C=C bonds are all terminal double bond, two of them being di-substituted, and one being the mono-substituted. Conformation analysis of β -elemene suggests these three double bond-containing substituents are likely located at an equatorial position in order to reach the lowest energy stage. It is clear to see that C-14 is slightly more steric hindered than C-13 due to its adjacency to C-15 and C-7 and C-8. Similarly, proton on C-2 is slightly more hindered than proton on C-4.

Selenium dioxide (SeO_2)-mediated allylic oxidation of olefin to allylic alcohol, commonly known as Riley oxidation, is one of the most important transformations in organic synthesis.²⁷ Typically, olefin was subjected to catalytic amount of SeO_2 and stoichiometric tert-butylhydroperoxide (TBHP) under mild condition. Since its discovery, Riley oxidation has been widely applied in numerous organic synthesis and natural product modifications.^{28, 29} The mechanism of Riley oxidation and the

DESCRIBED
THUS FAR

HEREIN WE REPORT
THAT SeO_2 CONDITIONS
CAN YIELD OTHER
OXIDATION
PATTERNS.

G-ROUND-
STATE

ALKENES
POSITIONED IN THE G-ROUND-STATE.
FURTHERMORE,
PROXIMITY
TO THE
QUATERNARY
CENTER.

1
2
3
4 preferences and selectivity of reaction sites of next to olefin functional group were
5 well documented in literature.^{30, 31} Theoretically, such transformation^S might access all
6 possible allylic hydrogen atoms (if any) with certain preference rule. When the olefin
7 substrate bears multiple C=C bonds, the reaction is expected to be less
8 region-selective, hence generating multiple products. In the case of β -elemene, there
9 are 4 different allylic protons namely protons at C-2, C-4, C-13, and C-14,
10 respectively. Our interests in modifying unexplored positions of β -elemene prompt us
11 to examine the SeO_2 -TBHP condition on this substrate. Of the all four
12 hydrogen-bearing allylic carbons, C-2 and C-4 draw our attention. We envision that
13 SeO_2 -mediated oxidation reaction might access the hydrogen atoms on these two
14 carbons, in addition to C-13 and C-14, hence, might install the hydroxyl group on
15 these two positions of cyclohexyl ring. As a result, the modification products of
16 β -elemene on its cyclohexyl ring could be obtained. Those modifications on
17 cyclohexyl ring of β -elemene represent the synthetic challenge up to date.
18 Additionally, SeO_2 -mediated allylic oxidation will also generate the oxidative
19 products from C-13 and C-14 (plus the possible combination). These products, though
20 previously reported, can only be synthesized in several steps.^{23, 24, 32-34}

Results

37
38 β -Elemene raw material is the free gift from Holley Kingkong Pharmaceutical Co.,
39 Ltd. GC-MS analysis³⁵ suggests that it contains only about 78% of β -isomer, plus the
40 other 3 isomers. Since the four elemene isomers process very similar structure and
41 physical properties, to purify them in the laboratory represents a challenge. Therefore,
42 the material was used as-is. It is understandable that the presence of other isomers will
43 generally give lower yield as well as complicate the isolation process.

44
45
46
47
48
49
50
51
52
53 β -Elemene raw material was subjected to a SeO_2 -mediated oxidation reaction in
54 CH_2Cl_2 , with 5 equivalent^S of TBHP at 0 °C for 6 hours. After standard workup
55 process, the crude product was purified in silica gel chromatography (petroleum ether
56 (PE)/ethyl acetate (EA)) to yield four fractions, with polarity from the least to the

THE TERM IS
"ALLYLIC"

NOT FOLLOWING,
"PREFERENCE" (REGIO - β CHEMOSELECTIVITY?)
WILL BE DETERMINED BY STEREOELECTRONICS.

RUN ON SENTENCE...
NOT VERY CLEAR.

most: fraction I ($R_f = 0.9$, PE/EA = 4:1, 7.4% yield), fraction II ($R_f = 0.7$, PE/EA = 4:1, 2.8% yield), fraction III ($R_f = 0.2$, PE/EA = 4:1, 8.6% yield), and fraction IV ($R_f = 0.15$, PE/EA = 4:1, 21.6% yield) (Scheme 1).

Scheme 1. Synthesis of β -elemene derivatives via SeO_2 -mediated oxidation reaction

The above four fractions were analyzed by HPLC. The results revealed that only fraction IV contains a single compound, the rest three fractions were all mixtures of two compounds.

Fraction I appears to be a single spot in TLC (petroleum-ethyl acetate system). After screening with several mix solvent systems for TLC, petroleum ether/acetone system was found to be the best solvent to resolve the two compounds (Scheme 2). The structure of compound **2** was established by NMR in comparison with literature reference.³³ The structure of compound **3** was elucidated through various 2D NMR techniques (see supporting information for details). It should be noted that compound **2** was synthesized in literature involving three steps and a tedious HPLC purification process.³³

Appendix B

Reviewers of Royal Society Open Science

March 14, 2020

Reference#: RSOS-200038

Dear Reviewers,

Thank you so much for your precious time reviewing our manuscript. The comments and suggestion are valuable, and will definitely help us improve the quality of our manuscript.

Below we would like to address the comments raised by you in point-to-point basis.

Reviewer # 1

1. "...On this line, a figure summarizing known semisynthesis and this report would be valuable".

Response: We have inserted a new figure (Figure 2) into manuscript per your suggestion.

2. "... The manuscript is very difficult to read. The document as a whole needs significant writing improvement".

Response: Thank you for your suggestion. We have revised our manuscript extensively. If there is any error, please let us know.

3. "...Journal abbreviations are often incorrect. For example, I don't know and could never find reference 9 (what journal is just "Chemistry?"). Ref. 20, 24, 27, 28, 29, 30, etc are all incorrect abbreviations. This made it so I couldn't easily copy and paste references to check for accuracy. I strongly recommend a more close analysis of references prior to submission as a courtesy to referees."

Response: We are sorry for not providing the correct reference format in our first submission. We have fixed the problem, including correct journal abbreviations, DOI (if there is any), article title etc. Ref 9 is a journal in Chinese. The journal name is correct as it is.

4. "...I could see a summary of the known elemene semisynthesis being a valuable figure for the paper. Consider adding one"

Response: This is great suggestion. We added Figure 2 as mentioned above.

5. "Elemene should not be capitalized when used in the middle of a sentence (do not capitalize names of natural products). This is inconsistent throughout the manuscript."

Response: Thank you for pointed out. We have made the correction accordingly.

6. "Consider revising certain figures for effective use of "white space" and aesthetics."

Response: Thank you for the great suggestion. Scheme 3 has been modified for this purpose.

7. "The NMRs look correct, though a few of the ¹³C NMRs could have better S/N ratios."

Response: We have added ^{13}C NMR data and HRMS data for all the new compounds.

Finally, we appreciate your 2-page corrections. We adopted them in this new version.

Reviewer # 2

1. "What is the status of the prepared compounds, when compared to standard drugs used for inhibition of proliferation of the cancer cell lines". "Table 1. Include IC 50 values of selected standard drugs for Inhibition of cell proliferation against A549 and U-87MG cell lines for comparison".

Response: In Table 1, the second row listed the activity of β -elemene. Its IC_{50} s are greater than 300 μM in A549 and U87 cell lines. Several analogs prepared in this manuscript have better activity than β -elemene. We always include STS in all of our experiment, just forgot to include in our first submission. In this revision, we have added STS data in the last row of Table 1. In addition, the standard deviation for each the assay were added.

2. "Include the IC_{50} values of other elemenes (alpha, gamma and delta) in Table 1."

Response: The separation of β -elemene from its regioisomers α -, γ -, δ - represents very tough task. That is why the pure form of β -elemene is very expensive and has very limited supply. To our knowledge, other isomers α -, γ -, δ - are not commercially available. This manuscript used elemene extract containing 78% β -isomer as starting material (as indicated in the manuscript). The major focus of this manuscript is to synthesize key intermediates or derivatives of β -elemene for biological evaluation and for the further analogs synthesis. Additional work to separate all isomers from elemene extract is ongoing in our lab. We will report the results once we have them.

3. "Authors may through light on the mechanism of improved action on the cell lines."

Response: In term of mechanism for improvement of activity, we postulate that the aldehyde functional group might pick up additional interactions with cell (such as hydrogen bonding), but at this moment we can't say it definitely. In page 10 first paragraph, we added a sentence "It is interesting to note that all the compounds showing better biological activities possess a aldehyde functional group. The reason for such phenomenon is currently being investigated in our laboratory."

4. "Whether highly active compound 17 (in the MS) is new? It may be highlighted in the abstract."

Response: Compound 17 is new compound not been reported before. We have included this point in Abstract. Thank you for great suggestion.

5. "Authors may include statistical analysis for the activity studies."

Response: We have revised Table 1 to include the SD and the appropriate caption below table.

6. "Conclusion-last sentences: Our approaches represent the first success to modify cyclohexyl ring, a challenging task unexplored before."

Response: We have revised the wording for the last sentence: “Our approaches represent the first success to introduce functional groups onto the cyclohexyl ring, a challenging task unexplored before”.

7. “¹H NMR and ¹³C NMR data for all the compounds may be presented in two tables (one for each ¹H and ¹³C) along with assignments (proton and carbon numbers)”

Response: Sorry, we did not adopt this suggestion, as the journal did not allow the tabulated NMR data (per Editor’s suggestion).

8. “The following references may be included at appropriate place in MS.”

Response: This is great suggestion. We added “Helvetica Chimica Acta 2004, 56(7):2270 – 2279” as Ref 27; and added “Molecules 2015, 20, 10205-10243” as Ref 29. The reference numbering was changed accordingly.

Thank you once again for your precious time! Should you have any question or suggestion, please contact us.

Sincerely yours,

Xiang-Yang Ye

Professor
Holistic Integrative Pharmacy Institutes (HIPI), School of Medicine
Hangzhou Normal University,
Hangzhou, Zhejiang 311121, China
Email: xyye@hznu.edu.cn
Tel: 86-571-28860236